# Ultraslow isomerization in photoexcited gas-phase carbon cluster $C_{10}^-$

K. Saha[1], V. Chandrasekaran [1,4], O. Heber[1], M.A. Iron [2], M.L. Rappaport[3] & D. Zajfman[1]

Isomerization and carbon chemistry in the gas phase are key processes in many scientific studies. Here we report on the isomerization process from linear $C_{10}^-$ to its monocyclic isomer. $C_{10}^-$ ions were trapped in an electrostatic ion beam trap and then excited with a laser pulse of precise energy. The neutral products formed upon photoexcitation were measured as a function of time after the laser pulse. It was found using a statistical model that, although the system is excited above its isomerization barrier energy, the actual isomerization from linear to monocyclic conformation takes place on a very long time scale of up to hundreds of microseconds. This finding may indicate a general phenomenon that can affect the interstellar medium chemistry of large molecule formation as well as other gas phase processes.

[1] Department of Particle Physics and Astrophysics, Weizmann Institute of Science, Rehovot 7610001, Israel. [2] Department of Chemical Research Support, Weizmann Institute of Science, Rehovot 7610001, Israel. [3] Department of Physics Core Facilities, Weizmann Institute of Science, Rehovot 7610001, Israel. [4] Present address: Department of Chemistry, School of Advanced Sciences, Vellore Institute of Technology, Vellore, 632014, India. Correspondence and requests for materials should be addressed to K.S. (email: koushik.saha@weizmann.ac.il) or to O.H. (email: oded.heber@weizmann.ac.il)

somerization of polyatomic systems, such as molecules and clusters, is a common phenomenon in nature. The importance of this process can be gauged by its impact on many fundamental and applied areas of research. From protein folding[1–3] to photosynthesis[4, 5], isomerization affects many basic biological processes. Various isomeric reactions are important in inorganic chemistry[6, 7] while cis-trans[8, 9] as well as other types of isomerizations[10] are vital in organic chemistry. Isomeric transitions have been utilized in such applied fields as sub-diffraction-limited lithography[11] and dye-sensitized solar cells[12]. Isomerization is also central in atmospheric chemistry[13] and in interstellar medium (ISM)[14]. In the case of ISM, one example of a key isomerization process is HCN ↔ NHC[15, 16], in which the line ratio of HCN/NHC is used to trace gas temperatures in galaxies and to compare their luminosities. Unlike in biological systems, where ambient factors like water and pH may dramatically affect the isomerization process[17, 18], this process in ISM occurs in the gas phase and hence depends solely on the internal molecular dynamics. Therefore, it is desirable to study such processes in gas phase in which less parameters are involved.

One of the most important observables that reflects the isomerization process is the isomerization rate. In the gas phase, it is usually assumed that isomerization rates are governed by molecular dynamics when there is enough internal energy to reach different isomeric paths. Typical rates have the same time constant as rotation, i.e., atomic motion relative to the center of mass, which is on the order of picoseconds. This was demonstrated in many experiments and calculations[19], although some faster[20] and slower processes[21] have also been measured.

Here, we present evidence of extremely slow isomerization rates for the transition from a gas phase linear $C_{10}^-$ cluster to its monocyclic conformation after photoexcitation above the isomerization barrier. When a polyatomic system such as $C_{10}^-$, which has a large number of degrees of freedom, is photoexcited, the excitation energy is quickly converted to vibrational energy due to nonadiabatic coupling between the electronic and vibrational degrees of freedom, a process known as internal conversion[22]. The system may then de-excite via various processes such as recurrent fluorescence[23], infrared emissions, fragmentation, or vibrational autodetachment, where neutrals are produced by the delayed detachment of electrons[24]. Depending on the internal energy of the excited system, neutrals are usually formed after a delay following photoexcitation. The neutralization rate is thus governed by the dynamics of the excited system and can reveal great insights into various de-excitation processes. In this study, we measure the neutralization rate of photoexcited linear $C_{10}^-$ anions at various photoexcitation energies ranging from below the isomerization barrier to above it. The contribution of isomerization to the neutralization rate is prominent and indicates very slow isomerization on the time scale of many microseconds. To the best of our knowledge, such a slow isomerization is observed for the first time in a polyatomic system in gas phase. Carbonaceous molecules and clusters are the most abundant polyatomic species in the ISM[25] and are proposed to be one of the main contributors to the unidentified infrared emission bands and diffused absorption bands[26]. Negative ions of such species, though predicted to be lower in abundance than their neutral counterparts, can play an important role in many astrophysical processes[27]. One of the most important creation mechanisms of carbonaceous species, assuming bottom-up carbon growth occurs, is probably the transition from 1D (linear carbon chains) to 2D and 3D structures. Hence, our results may shed light on the more general unexplained creation of large carbon-based systems like polycyclic aromatic hydrocarbons (PAH)[28] and $C_{60}$[29, 30] in the ISM. Furthermore, our results on isomerization from linear to

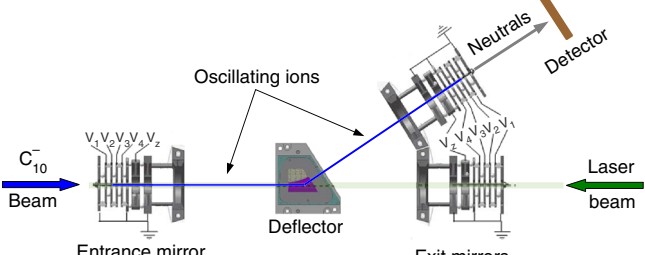

**Fig. 1** Schematic of the experimental setup. The $C_{10}^-$ beam is trapped oscillating between the entrance mirror and the exit mirror in the bent section of the electrostatic ion beam trap (EIBT) by applying a specific set of voltages to the mirror electrodes and to the electrostatic deflector. A laser pulse of ~5 ns duration is admitted through the linear section of the EIBT and merged with the trapped $C_{10}^-$ ions. Neutrals produced promptly after laser interaction fly out of the setup from the linear section and are not detected. Neutrals produced after the ions are deflected towards the bent section by the deflector, i.e., after a delay of at least a few microseconds after the interaction with the laser, fly out of the bent section exit mirror and are detected by a microchannel plate (MCP) detector and counted as a function of trapping time

cyclic $C_{10}^-$ may also be instructive in nanographene production studies[31].

## Results

**Experimental scheme.** Vibrationally and rotationally excited $C_{10}^-$ is produced by a cesium sputter source and trapped in a bent electrostatic ion beam trap (EIBT)[32] (shown schematically in Fig. 1). In the EIBT, by applying appropriate voltages to the entrance and exit mirrors, an ion beam can be kept oscillating in a field-free region between them, thus trapping the ions for a certain duration. In this study, the $C_{10}^-$ beam was trapped for 250 ms. The trapped ion beam was merged with a single 5 ns-long pulse of photons from an energy-tunable laser after about 190 ms of trapping (see Supplementary Note 1 and Supplementary Fig. 1 for discussion about photoexcitation at shorter trapping times than 190 ms). The neutral species flying out of the bent section exit mirror, produced by electron detachment from $C_{10}^-$ either due to the laser impact or by residual gas collisions, are detected by a microchannel plate (MCP) detector with a phosphor screen and recorded as a function of trapping time. A typical neutral counts versus trapping time curve is shown in Fig. 2.

The peak at ~190 ms is due to photon-induced delayed detachment. The peak is magnified and shown in the inset of the figure. Any neutrals that are produced promptly after the laser interaction are not recorded by the detector since they fly out of the linear section of EIBT. Thus, the neutral counts shown in the inset of Fig. 2 are only those that are produced by vibrational autodetachment after a significant delay (>few microseconds) from the time of laser impact. It can be clearly seen that the neutrals are produced even after a few hundred microseconds, and it is this neutralization rate that we studied to understand the dynamics of the excited system.

**Neutralization of $C_{10}^-$ at different photon energies.** $C_{10}^-$ is the smallest carbon cluster anion for which the existence of two isomeric forms (linear and monocyclic) has been experimentally reported[33]. The theoretical adiabatic electron affinity (AEA) and experimental electron affinity of linear $C_{10}^-$ are 4.22[34] and 4.46 eV[35], respectively, while those of monocyclic $C_{10}^-$ are 1.66[34] and 2.3 eV[36], respectively. The isomerization barrier energy ($E_B$) was determined by generating the potential energy surfaces using density functional theory (DFT) (shown schematically in Fig. 3)

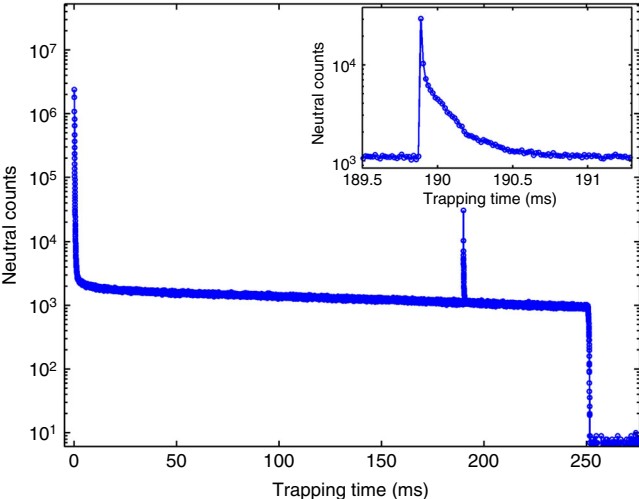

**Fig. 2** Neutralized $C_{10}^-$ counts as a function of trapping time. Ions were trapped and a laser pulse of 2.85 eV photons was admitted ~190 ms afterwards. Trapping was stopped after 250 ms and the cycle repeated. The data shown is from ~$10^5$ such trapping cycles. The high number of counts during the first few milliseconds is due to spontaneous electron emission from the vibrationally and rotationally hot $C_{10}^-$. During this time, $C_{10}^-$ also undergoes radiative cooling and spontaneous electron emission stops. The neutral counts after this time are due to loss of an electron upon collision of the trapped ions with the residual gas. The counts after 250 ms are due to MCP noise. The peak at ~190 ms is produced by neutrals formed due to delayed detachment of electrons. The inset shows an enlargement of this peak, which clearly depicts formation of neutrals with significant delay relative to the laser interaction time

and calculated to be 2.68 eV above the linear $C_{10}^-$ ground state. Since the trapped $C_{10}^-$ is rotationally excited, due to conservation of rotational angular momentum, the isomerization barrier, as well as all the energy levels of cyclic $C_{10}^-$, will be shifted to higher energies relative to the linear conformation[37]. The degree of this shift depends on the rotational temperature of the cluster at the time of the laser interaction.

The experiments were performed using several laser photon energies. Figure 4 shows the neutral counts as a function of time after interaction of $C_{10}^-$ with laser pulses of different energies. Neutrals are produced via photon-induced delayed electron emission upon vibrational autodetachment in anionic clusters and large molecular anions under certain conditions on the photoexcitation energy. For electron emission to occur, the photon energy must be sufficiently high such that the total internal energy of the system after photoexcitation is more than the adiabatic electron affinity. The time required for electron emission from such an anionic species depends on its internal dynamics and decreases with the increase in its total internal energy. Thus, if the photoexcitation energy is much higher than the adiabatic electron affinity, electrons are mostly emitted promptly after interaction with the photon. Electrons are emitted with a significant delay after photoexcitation only when the energies are close to the adiabatic electron affinity of a particular anionic species. The neutral counts observed more than a few microseconds after the laser pulse cannot be from monocyclic $C_{10}^-$ as the photon energies are much higher than the adiabatic electron affinity (also the experimental electron affinity) of this species. In such a scenario, the total internal energy of the system after photoexcitation would be very high. At these energies, electron detachment from monocyclic $C_{10}^-$ would be prompt and most of the neutrals produced would escape from the linear

section of the trap. We measured the neutrals produced immediately after the laser interaction by a detector placed after the linear section exit mirror (not shown in Fig. 1). The number of such neutrals was found to be negligible at these photon energies, indicating the absence of a significant amount of monocyclic $C_{10}^-$ in the trapped ion beam (see Supplementary Note 2 and Supplementary Fig. 2). Thus, the neutral counts we measure in our detector at the end of the bent section of the trap for hundreds of microseconds are from linear $C_{10}^-$. These neutrals also cannot be due to vibrational autodetachment of linear $C_{10}^-$ as the energies are much lower than the adiabatic electron affinity of the system. Hence, the total internal energy of linear $C_{10}^-$ after photoexcitation at these energies will not be sufficient for electron detachment.

To understand the long neutralization process lasting several microseconds, we calculated the neutralization rate using a statistical model that takes into account all the de-excitation processes occurring in photoexcited linear $C_{10}^-$. The results from the model for each photon energy are depicted by the lines in Fig. 4. For all photon energies, there is excellent agreement within experimental error between the neutralization rates computed by the model and those measured experimentally. The details of the model are discussed in the next section.

**Analysis of neutralization rate.** Photoexcited linear $C_{10}^-$ can relax by various complementary as well as competitive processes. The radiative relaxation mechanisms are infrared (IR) emissions upon transitions between the vibrational levels of the system, and recurrent fluorescence (RF) due to transitions between the electronic excited states[23, 38, 39]. The non-radiative de-excitation in the excited cluster can occur via vibrational autodetachment (VAD), which leads to electron detachment, or fragmentation. The lowest energy at which linear $C_{10}^-$ fragments is 5.68 eV[34], which is much higher than the excitation energies at which the experiments were performed. There might be the possibility of fragmentation upon two photon excitation, but it will not contribute to the slow production of neutrals. In light of this, we have neglected fragmentation as a viable relaxation mechanism in our analysis. All these processes occur in competition with each other with different rates that depend on the total energy of the excited system. These processes are common in any $C_n^-$, where $3 < n < 10$. However, an additional process, isomerization, can occur in competition with these processes when $n \geq 10$[33]. In our case, it is the isomerization from linear to monocyclic $C_{10}^-$. After isomerization, the prominent relaxation mechanisms at these excitation energies for monocyclic $C_{10}^-$ are IR emissions and fast electron detachment by VAD, producing neutrals. Thus, the neutrals that are produced in our experiment can be either due to VAD directly from linear $C_{10}^-$ or fast VAD from monocyclic $C_{10}^-$ after isomerization. With the excitation energies used in our experiments, the latter process is more likely, and this is confirmed by our statistical model calculations.

The neutralization rate is determined using statistical rate theory[40] considering the contributions from all relaxation mechanisms. The relaxation mechanisms are governed by their rate coefficients, which depend on the internal energy of the system, $E$, after photoexcitation. The rate coefficients for the VAD, IR and RF emission processes are estimated by statistical phase space theory[41–43]. The rate coefficient for VAD, based on a detailed balance approach considering the Langevin cross section for electron capture[44, 45], is approximated by[43]

$$k_{VAD_{l,c}}(E) = f_{VAD_{l,c}} \int_0^{E-AEA_{l,c}} \frac{2\mu G_{l,c} e}{\pi \hbar^3} \sqrt{2\alpha_{l,c}} \sqrt{\varepsilon} \frac{\rho_{l,c}(E - AEA_{l,c} - \varepsilon)}{\rho_{l,c}^-(E)} d\varepsilon$$

(1)

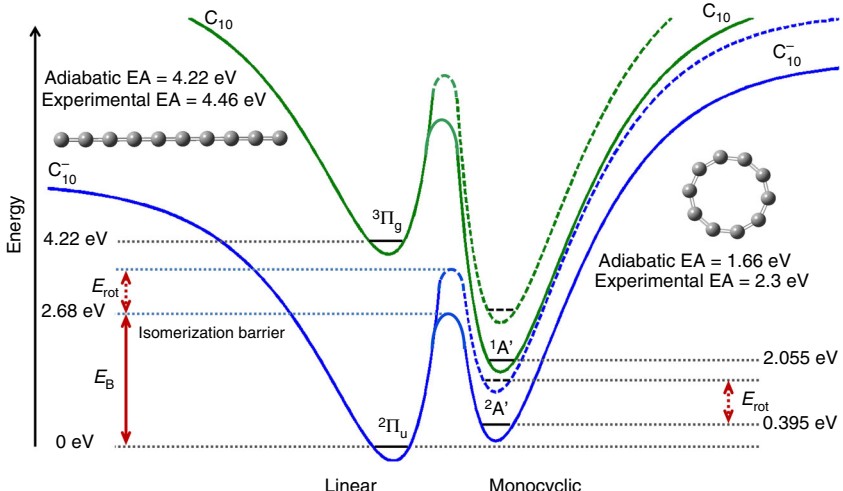

**Fig. 3** Schematic of the DFT potential energy surfaces of $C_{10}^-$. The ground states of linear and monocyclic $C_{10}^-$ are shown by the solid lines. The energy shift due to rotational energy of the linear $C_{10}^-$ is set to 0 eV in the energy scale. The energy levels for monocyclic $C_{10}^-$ are shown relative to the linear $C_{10}^-$ energy scale. Due to conservation of rotational angular momentum during isomerization from linear to monocyclic $C_{10}^-$, the isomerization barrier as well as the energy levels of monocyclic $C_{10}^-$ will be shifted with respect to linear $C_{10}^-$ (shown by dashed lines). $E_{rot}$ is the amount by which this shift occurs and is the difference of rotational energies between the linear and monocyclic $C_{10}^-$ with a particular rotational quantum number. The probability that the system is in a certain rotational state (with a particular quantum number) depends on the rotational temperature of the cluster at the time of laser interaction

where the indices $l$ and $c$ denote the parameters for linear and monocyclic conformations, respectively, $\mu$ is the reduced electron mass, $G$ is the ratio of neutral-to-anion electronic degrees of freedom ($G_l = 3/2$ and $G_c = 1/2$), $\varepsilon$ and $e$ are the electronic energy and charge, respectively, and $\alpha$ is the polarizability of the neutrals (for linear $C_{10}$, $\alpha_l = 300.32$ a.u[46]. and for monocyclic $C_{10}$, $\alpha_c = 108.65$ a.u.[47]). $\rho$ and $\rho^-$ are the densities of the vibrational states of $C_{10}$ and $C_{10}^-$, respectively, in their electronic ground states calculated using the harmonic oscillator approach and the Beyer–Swinehart method[48]. The fundamental vibrational frequencies used to calculate the densities of states were obtained from our DFT calculations. $f_{\mathrm{VAD}_{l,c}}$ is a correction factor we introduced to account for any possible shortcomings in our model as determined by the experiment results.

The rate coefficient for IR radiative relaxation is determined by[43]

$$k_{\mathrm{IR}_{l,c}}(E) = f_{\mathrm{IR}_{l,c}} \sum_s A_s(h\nu_s) \sum_{n \geq 1} \frac{\rho_{l,c}^-(E - nh\nu_s)}{\rho_{l,c}^-(E)} \qquad (2)$$

The harmonic frequencies $\nu$ and their corresponding IR intensities used to calculate the Einstein coefficient $A_s(h\nu_s)$ for IR transitions were obtained from our DFT calculations. The ratio of the densities of states $\rho_{l,c}^-$ gives the probability that $C_{10}^-$ will be in a state that carries $n$ vibrational quanta $h\nu$. $f_{\mathrm{IR}_{l,c}}$ is the correction factor to compensate for any drawbacks in our model as determined by the experimental data.

The rate coefficient for RF radiative relaxation is computed by[43]

$$k_{\mathrm{RF}_l}(E) = f_{\mathrm{RF}} \sum_{j = 1 - 4} A_j(E_j) \frac{\rho_l^-(E - E_j)}{\rho_l^-(E)} \qquad (3)$$

where $A_j(E_j)$ denotes the Einstein coefficient for transition from electronic state $j$ to the $^2\Pi_u$ ground state of linear $C_{10}^-$ and are computed by considering the electronic states and oscillator strengths as per CASPT2/cc-pVTZ//CCSD(T)/6-31G(d) results

reported in ref. [49]. The ratio of the density of states, $\rho_l^-$, in Eq. (3), gives the probability that linear $C_{10}^-$ will be excited to a particular electronic state. The correction factor introduced to overcome any shortcomings in the model as determined by the experimental data is denoted by $f_{\mathrm{RF}}$. We do not considered RF to be a major relaxation mechanism in monocyclic $C_{10}^-$ due to the absence of any known excited electronic state close to its ground state.

Finally, the rate coefficient for isomerization from linear to monocyclic $C_{10}^-$ is obtained by Rice-Ramsperger-Kassel-Marcus (RRKM) theory and is given by[14, 37]

$$k_{\mathrm{iso}}(E) = f_{\mathrm{iso}} \frac{N_{\mathrm{ts}}(E - E_{\mathrm{B}})c}{\rho_l^-(E)} \qquad (4)$$

where $N_{\mathrm{ts}}(E - E_{\mathrm{B}})$ is the sum of the vibrational states in the transition state that have energy greater than or equal to the isomerization barrier energy $E_{\mathrm{B}}$ and is estimated by the Beyer–Swinehart method[48] using the vibrational frequencies of the transition state obtained from our DFT calculations. $c$ is the speed of light and $f_{\mathrm{iso}}$ is the correction factor as determined by experimental results. During isomerization, the rotational angular momentum of the systems is conserved. Due to different symmetries and rotational constants of linear and monocyclic $C_{10}^-$[50], the shifts in their energy levels due to rotational energy are different. Thus, $E_{\mathrm{B}}$ is also shifted by an amount equal to the difference in the rotational energies of the two isomers. For our model, we assume an initial population before laser interaction that follows a Boltzmann distribution at a rotational temperature of 1000 K. The calculations were performed for eight different rotational quantum numbers spanning this initial population distribution. Varying the rotational temperature by ±500 K and increasing the number of rotational quantum numbers in our calculations did not significantly affect the results. The degree that $E_{\mathrm{B}}$ shifts in energy is different for each rotational quantum number, and the contribution to the isomerization rate for each quantum number is accounted for in the calculations.

The total neutralization rate $\mathcal{R}(t)$ due to one- and two-photon processes taking into account all of the relaxation mechanisms

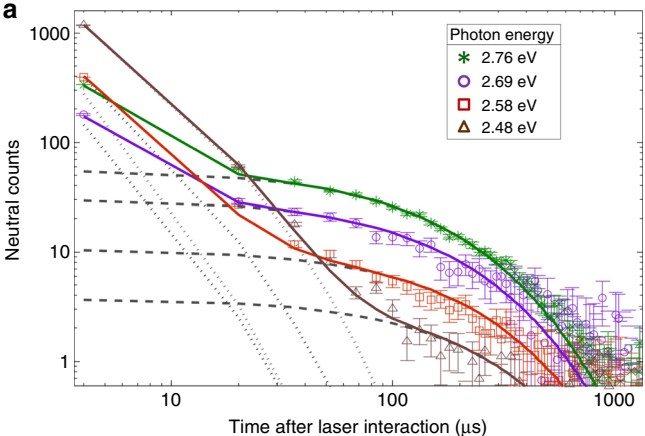

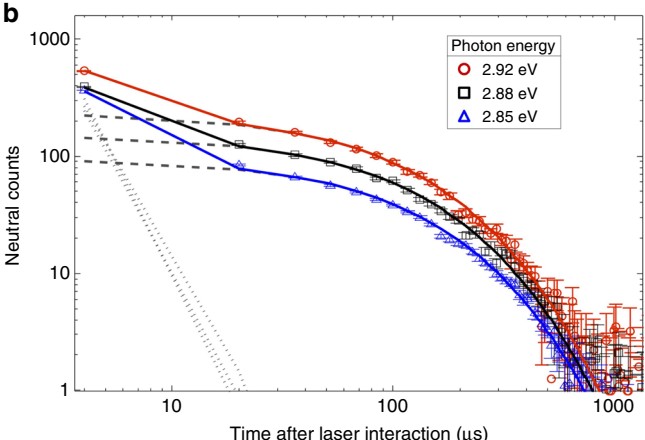

**Fig. 4** Normalized neutral counts as a function of time after laser interaction with $C_{10}^-$. **a** Photoexcitation energies from below to just above the isomerization barrier, **b** photoexcitation energies well above the barrier. The results shown here for each photon energy were obtained by collecting data for ~$10^5$ trapping cycles. The zero of the time axis is the time of laser interaction. The contribution of neutrals produced due to residual gas collisions has been subtracted and the data normalized with respect to background and laser photon counts. The symbols are the experimental data. The error bars are the standard deviation of the experimental data. The lines are from theoretical calculations using the model. The dashed lines denote contributions from one-photon excitations while the dotted lines are due to excitations from two photons. The solid lines represent the contribution from both and agree well with the experimental data

can be calculated using Eqs. (1), (2), (3), and (4). The neutralization rate $\mathcal{R}_1(t)$ by VAD directly from linear $C_{10}^-$ is given by[39]

$$\mathcal{R}_{1_{1,2}}(t) = N_{1,2} \int_{\mathrm{AEA_l}}^{\infty} f_0\Big(E = E_\mathrm{i} + E_{\mathrm{phot}_{1,2}}\Big) k_{\mathrm{VAD_l}}(E)\, e^{-k_{\mathrm{tot_l}}(E)t}\, \mathrm{d}E \quad (5)$$

where the indices 1, 2 denote one- and two-photon excitation processes and $N$ is the corresponding normalization factor. $f_0\big(E = E_\mathrm{i} + E_{\mathrm{phot}_{1,2}}\big)$ is the internal energy population distribution of the laser excited linear $C_{10}^-$, where $f_0(E = E_\mathrm{i})$ denotes the initial internal energy population distribution before laser interaction. The vibrational temperature used to estimate the initial internal energy population distribution was set to 670 K as this yielded the best agreement with the experimental data. $k_{\mathrm{tot_l}}(E) = k_{\mathrm{VAD_l}}(E) + k_{\mathrm{IR_l}}(E) + k_{\mathrm{RF_l}}(E) + k_{\mathrm{iso}}(E)$ gives the total de-excitation rate considering all relaxation mechanisms of the linear isomer.

Neutrals are produced upon isomerization by a consecutive two-step process. In the first step, linear

$C_{10}^-$ isomerizes to monocyclic $C_{10}^-$, and neutrals are then formed by VAD from monocyclic $C_{10}^-$ in the second step. The rate of production of neutrals in this case can thus be represented by[51]

$$\mathcal{R}_{\mathrm{iso}_{1,2}}(t) = N_{1,2} \int_{E_\mathrm{B}}^{\infty} f_0\Big(E = E_\mathrm{i} + E_{\mathrm{phot}_{1,2}}\Big) \times$$
$$\frac{k_{\mathrm{iso}}(E) k_{\mathrm{VAD_c}}(E)}{k_{\mathrm{tot_c}}(E) - k_{\mathrm{tot_l}}(E)} \Big(e^{-k_{\mathrm{tot_l}}(E)t} - e^{-k_{\mathrm{tot_c}}(E)t}\Big)\, \mathrm{d}E \quad (6)$$

where $k_{\mathrm{tot_c}}(E) = k_{\mathrm{VAD_c}}(E) + k_{\mathrm{IR_c}}(E)$ gives the total rate for the relaxation mechanisms of monocyclic $C_{10}^-$.

The neutralization rate for one-photon excitations is represented by

$$\mathcal{R}_1(t) = \mathcal{R}_{\mathrm{iso}_1}(t) + \mathcal{R}_{1_1}(t) \quad (7)$$

while that due to two-photon excitations is denoted by

$$\mathcal{R}_2(t) = \mathcal{R}_{\mathrm{iso}_2}(t) + \mathcal{R}_{1_2}(t) \quad (8)$$

The total neutralization rate is thus is given by

$$\mathcal{R}(t) = \mathcal{R}_1(t) + \mathcal{R}_2(t) \quad (9)$$

The total neutralization rate calculated by our model represents the neutralization rate that we measure in our experiment for various photon energies.

## Discussion

The total neutralization rate $\mathcal{R}(t)$ calculated by our model for each photon energy is depicted in Fig. 4 by solid lines and shows very good agreement with the experimental data. The correction factors for all the rate coefficients were set to unity except for $f_{\mathrm{RF}}$ which was set to 0.8 so that the calculated neutralization rates best match our experimental results. The contribution to the neutralization rate from the one-photon excitations $\mathcal{R}_1(t)$ are shown by dashed lines in Fig. 4 while those for the two-photon excitations $\mathcal{R}_2(t)$ are depicted by dotted lines. As expected for these photon energies, we observed that VAD directly from linear $C_{10}^-$ does not contribute to the one-photon neutralization rate, i.e., $\mathcal{R}_1(t) = \mathcal{R}_{\mathrm{iso}_1}(t)$, and it is this neutralization pathway that occurs over an unusually long time of up to hundreds of microseconds. At a photon energy of 2.48 eV, the neutralization is dominated by two-photon processes with very little contribution from one-photon excitation, because very little of the initial population distribution is excited above the isomerization barrier energy by absorbing a single photon. However, in a two-photon process, there is more than sufficient energy to overcome the barrier for isomerization, and thus we see faster neutralization rates as the rate coefficients are much larger at higher energies. As the photon energy increases, more of the initial population distribution is excited above the isomerization barrier by one-photon processes, resulting in an increased contribution to the total neutralization rate, and this becomes dominant at energies higher than the isomerization barrier. It should be noted that isomerization starts at a photon energy of 2.58 eV, which is lower than the isomerization barrier energy because the linear $C_{10}^-$ in our experiments is not in the ro-vibrational ground state when photoexcited. This initial internal energy of the system coupled with the photoexcitation energy is sufficient for part of the population to cross the isomerization barrier.

The rate coefficients for each relaxation mechanism are plotted in Fig. 5. The rate coefficients for isomerization and for relaxations from monocyclic $C_{10}^-$ in Fig. 5 include the effect of the rotational energy shift corresponding to the smallest rotational quantum number considered for calculations. Due to the high rate coefficient for VAD from monocyclic $C_{10}^-$, neutrals will be produced by this process without any significant time delay.

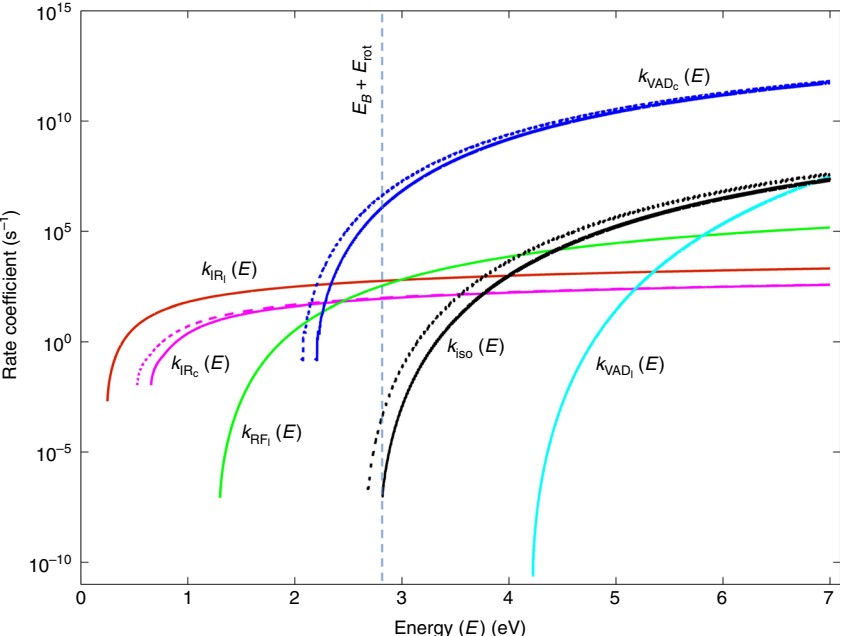

**Fig. 5** Rate coefficients of the various relaxation mechanisms as a function of the internal energy ($E$). Energies are relative to the energy of linear $C_{10}^-$ after setting the shift due to its rotational energy to 0 eV. All rate coefficients are relative to this energy scale. The rate coefficients for isomerization and for relaxations from monocyclic $C_{10}^-$ are shown taking into account the rotational energy shift corresponding to the smallest rotational quantum number considered for calculations. The vertical line shows this shift for the isomerization barrier. Linear $C_{10}^-$ with internal energy above this barrier can undergo isomerization. The dashed lines are the rate coefficients for isomerization and for relaxations from monocyclic $C_{10}^-$ without considering any rotational energy shift. For the sake of completeness, the rate coefficients for VAD ($k_{VAD_c}(E)$) and IR ($k_{IR_c}(E)$) from monocyclic $C_{10}^-$ are shown even below the isomerization barrier energy though they come into play only after isomerization. The small rate coefficient for isomerization, $k_{iso}(E)$, evinces a slow isomerization rate

The slow rate for production of neutral species observed in our experimental results can only be explained by the small isomerization rate coefficient. Thus, the neutrals produced up to hundreds of microseconds after photoexcitation are due to slow isomerization from linear $C_{10}^-$ to monocyclic $C_{10}^-$ followed by immediate electron detachment by VAD from monocyclic $C_{10}^-$.

To the best of our knowledge, this is the first observation of an ultraslow isomerization in a cluster or molecule in the gas phase lasting hundreds of microseconds. Electron detachment mediated by isomerization is also observed in this study. Although in ISM the relative abundance of carbon cluster anions such as $C_{10}^-$ is predicted to be lower than their corresponding neutral counterparts, they can facilitate the formation of many neutral carbonaceous molecules, resulting in an enhancement of neutral abundances in environments such as dark molecular clouds[52]. This increase in neutral abundance is more prominent in reactions involving $C_{10}^-$ than with smaller anionic species. To discern all of the possible chemical pathways that can lead to the formation of new species in ISM requires understanding and characterization of all the competing processes and associated molecular parameters, like the ones discussed in this study. The ultraslow isomerization rates that we observe, along with the rates for other relaxation mechanisms in gas phase $C_{10}^-$, can be key inputs in astrochemical models that try to explain or predict the formation of various species in ISM. Our results, showing ultraslow isomerization in gas phase $C_{10}^-$, may be a general phenomenon in other large clusters or molecules and may help understand the dynamics of large molecular systems in the excited state.

## Methods

**Experimental details.** Carbon cluster anions were produced in a cesium sputter source by sputtering a graphite target with $Cs^+$ ions. Anions were then extracted and accelerated to 4.2 keV by applying the appropriate voltages to extractor and accelerator electrodes situated just after the source. The anion beam was chopped into 55 μs-long bunches by an electrostatic chopper. $C_{10}^-$ ions were separated from this anion bunch by mass selection in a magnetic field of a 90° bending magnet. The anion bunch was guided into the EIBT (Fig. 1) by various electrostatic deflectors, Einzel lenses and a quadrupole triplet. The pressure in the EIBT chamber was about $2 \times 10^{-10}$ Torr during the experiments. $C_{10}^-$ ions were trapped in the EIBT by applying the following voltages to its entrance and exit mirror electrodes: $V_1 = -6.75$ kV, $V_2 = -4.875$ kV, $V_3 = -3.25$ kV, $V_4 = -1.625$ kV, and $V_z = -4.10$ kV. To trap the ions in the bent section of the EIBT, ±670 V was applied to the electrodes of the spherical deflector. At the beginning of each trapping cycle, the voltages on the entrance mirror were lowered to 0 V in order to inject the ion bunch into the trap. Once the ion bunch was inside the EIBT, the entrance mirror voltages were quickly (~100 ns) raised before the ions returned after being reflected from the exit mirror. The $C_{10}^-$ ions were thus trapped in an oscillating trajectory in the field free region between the two mirrors. The oscillation period of the $C_{10}^-$ ions in the EIBT was 16 μs.

The gain of the MCP detector placed at the end of the bent section exit mirror was set at about 2 kV, while that of the phosphor screen placed behind it was kept at about 4 kV to detect the neutrals from $C_{10}^-$. The detector signal was amplified by a preamplifier and processed by a constant fraction discriminator before being digitized and recorded as a function of time by a National Instruments 6602 timer/counter multiscalar card installed in a computer using a 20 MHz clock (50 ns time resolution). LabVIEW[53] programs were used to control the experimental parameters and to monitor and store the data in a list-mode file on an event-by-event basis.

**DFT calculations.** The isomerization barrier energy and vibrational frequencies of the transition state as well as of the linear and monocyclic $C_{10}^-$ and $C_{10}$ along with their corresponding IR intensities were calculated by density functional theory (DFT)[54]. The calculations were performed using Gaussian09 Revision D.01[55] and ORCA version 3.0.3[56]. Gaussian09 was used for all calculations other than the double-hybrid energy calculations, for which ORCA was used. The calculations were done by optimizing the geometries for linear and monocyclic $C_{10}^-$ and $C_{10}$ and the connecting transition states. Geometries were optimized with Adamo and

**Table 1 Vibrational frequencies ($\nu$, cm$^{-1}$) and IR intensities (km mol$^{-1}$) of the linear and monocyclic C$_{10}^{-}$ and vibrational frequencies ($\nu$, cm$^{-1}$) of the transition state and linear and monocyclic C$_{10}$**

| Linear C$_{10}^{-}$ | | Monocyclic C$_{10}^{-}$ | | Transition state | Linear C$_{10}$ | Monocyclic C$_{10}$ |
|---|---|---|---|---|---|---|
| $\nu$ | IR intensity | $\nu$ | IR intensity | $\nu$ | $\nu$ | $\nu$ |
| 43.605 | 14.553 | 138.004 | 0.249 | −247.523 | 40.780 | 181.708 |
| 44.006 | 13.072 | 158.096 | 3.312 | 76.152 | 40.780 | 181.709 |
| 111.811 | 0.000 | 213.593 | 1.714 | 114.431 | 104.723 | 235.480 |
| 114.446 | 0.000 | 240.287 | 0.000 | 151.151 | 104.723 | 235.480 |
| 198.057 | 15.417 | 355.499 | 7.992 | 181.313 | 182.211 | 436.265 |
| 207.335 | 9.534 | 374.256 | 59.051 | 248.411 | 182.211 | 454.220 |
| 279.190 | 0.000 | 378.483 | 7.721 | 250.937 | 259.634 | 454.220 |
| 279.737 | 0.000 | 467.656 | 3.876 | 381.122 | 259.634 | 455.301 |
| 381.226 | 2.390 | 473.347 | 0.000 | 400.399 | 370.731 | 498.482 |
| 400.609 | 0.586 | 479.978 | 57.232 | 407.459 | 370.731 | 498.482 |
| 412.310 | 0.000 | 491.871 | 0.000 | 453.680 | 419.938 | 510.200 |
| 474.943 | 0.000 | 499.407 | 0.246 | 466.981 | 479.901 | 510.200 |
| 501.504 | 0.000 | 509.766 | 0.049 | 486.891 | 479.901 | 568.093 |
| 512.563 | 2.462 | 575.082 | 37.140 | 499.973 | 524.899 | 568.093 |
| 514.031 | 0.000 | 823.026 | 4.011 | 565.545 | 524.899 | 856.311 |
| 549.732 | 0.992 | 958.910 | 72.907 | 792.578 | 537.645 | 1081.477 |
| 563.845 | 0.000 | 1057.318 | 99.878 | 926.220 | 537.645 | 1081.477 |
| 796.017 | 41.744 | 1351.854 | 0.064 | 1199.552 | 810.449 | 1482.270 |
| 1154.220 | 0.000 | 1434.570 | 9.352 | 1392.925 | 1173.219 | 1506.370 |
| 1480.829 | 49.127 | 1530.446 | 32.001 | 1783.915 | 1506.340 | 1506.370 |
| 1912.916 | 0.000 | 1704.052 | 10.765 | 1882.131 | 1811.578 | 1962.315 |
| 2003.757 | 2351.481 | 1955.243 | 226.385 | 1918.078 | 2004.072 | 1962.315 |
| 2114.309 | 0.000 | 1998.975 | 315.677 | 2038.974 | 2086.088 | 2062.828 |
| 2179.331 | 0.000 | 2102.314 | 267.150 | 2110.288 | 2173.576 | 2062.829 |
| 2214.838 | 3382.104 | | | | 2207.694 | |

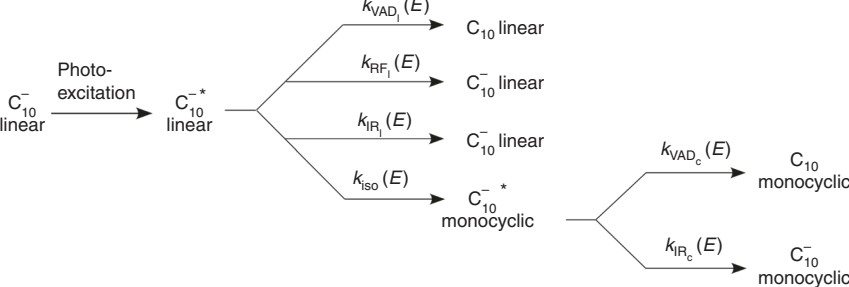

**Fig. 6** The relaxation mechanisms that occur upon photoexcitation of linear C$_{10}^{-}$. An asterisk indicates the species that are in an excited state

Barone's hybrid version (i.e., PBE0)[57] of the Perdew–Burke–Ernzerhof (PBE)[58] generalized-gradient approximation (GGA) functional with an added empirical dispersion correction[59], specifically the third version of Grimme's empirical dispersion correction[60, 61] with Becke-Johnson dampening[61–63]. The def2-TZVPPD (triple-$\zeta$ quality) basis set[64, 65] was used for geometry optimization calculations.

Energies were calculated using a Kozuch and Martin's dispersion-corrected (D3BJ), spin component scaled (i.e., an SCS[66, 67]-MP2[68]-like correlation contribution), double hybrid (DSD) functional, specifically DSD-PBEP86[69]. This functional incorporates the PBE exchange[58] and the Perdew-86 (P86) correlation[70] functionals. The def2-QZVPPD (quadruple-$\zeta$ quality) basis set[65] was used for the double hybrid energy calculations. The efficiency of the calculation was improved by using the resolution of identity-chain of spheres (RIJCOSX) approximation[71]. This class of DFT functionals has shown to provide energies approaching that of the "Gold Standard" in computational chemistry. There are a number of reviews and benchmark studies of double-hybrid functionals that clearly show that the use of this class of functionals is highly recommended[72]. Energies reported include zero-point vibrational energy (ZPVE) corrections calculated at the PBE0/def2TZVPPD level of theory. Frequencies were scaled by 0.9824 as recommended by Kesharwani et al.[73]. The calculated vibrational frequencies of linear and monocyclic C$_{10}^{-}$ and their corresponding IR intensities are given in Table 1 along with the vibrational frequencies of the transition state and linear and monocyclic C$_{10}$.

**Neutralization rate upon isomerization**. The neutralization rate and the rate coefficients were computed by a statistical model using MATLAB[74]. The functional

form of the neutralization rate upon isomerization, i.e., $\mathcal{R}_{iso}(t)$, was obtained by considering the rate equations for the parallel and consecutive reactions for photoexcited linear C$_{10}^{-}$ as shown in Fig. 6. Isomerization from excited linear C$_{10}^{-}$ occurs in competition with other relaxation mechanisms that take place in parallel. Neutral species are produced upon isomerization by a consecutive reaction where linear C$_{10}^{-}$ isomerizes to excited monocyclic C$_{10}^{-}$, that then undergoes VAD to form monocyclic C$_{10}$, which is in competition with simultaneous IR relaxation. Though isomerization may be a reversible process, for the sake of simplicity and considering the high rate coefficient for VAD from monocyclic C$_{10}^{-}$ we neglect this possibility in our analysis.

In a consecutive reaction of the form

$$\mathcal{A} \xrightarrow{k_a} \mathcal{B} \overset{k_{b1}}{\underset{k_{b2}}{\rightrightarrows}} \begin{matrix} \mathcal{C} \\ \mathcal{D} \end{matrix},$$

the rate equation for $\mathcal{B}$ is given by

$$\frac{d[\mathcal{B}]}{dt} = k_a[\mathcal{A}] - k_{b1}[\mathcal{B}] - k_{b2}[\mathcal{B}] \tag{10}$$

where $k_a$ is the rate of creation of $\mathcal{B}$ while $k_{b1}$ and $k_{b2}$ are the rates of depletion of $\mathcal{B}$. $[\mathcal{A}]$ and $[\mathcal{B}]$ are the concentrations of $\mathcal{A}$ and $\mathcal{B}$, respectively.

In the formation of neutrals via isomerization, $\mathcal{A}$ is the photoexcited linear C$_{10}^{-}$, the intermediate state $\mathcal{B}$ is the excited monocyclic C$_{10}^{-}$, $\mathcal{C}$ is the end product, i.e., monocyclic C$_{10}$ and $\mathcal{D}$ is the side product, monocyclic C$_{10}^{-}$. The intermediate state

$\mathcal{B}$ is created by isomerization, thus

$$k_a[\mathcal{A}] = k_{\text{iso}}(E)f_0(E)e^{-k_{\text{tot}_l}(E)t}$$

considering the fact that isomerization occurs in parallel with other relaxation processes from excited linear $C_{10}^-$. The intermediate excited monocyclic $C_{10}^-$ is depleted by VAD and IR relaxation processes; thus

$$k_{b1}[\mathcal{B}] + k_{b2}[\mathcal{B}] = k_{\text{VAD}_c}(E)[\mathcal{B}] + k_{\text{IR}_c}(E)[\mathcal{B}] = k_{\text{tot}_c}(E)[\mathcal{B}]$$

Therefore, Eq. (10) becomes

$$\frac{d[\mathcal{B}]}{dt} = k_{\text{iso}}(E)f_0(E)e^{-k_{\text{tot}_l}(E)t} - k_{\text{tot}_c}(E)[\mathcal{B}] \qquad (11)$$

$[\mathcal{B}]$ can be derived by solving this differential equation, yielding

$$[\mathcal{B}] = f_0(E)\frac{k_{\text{iso}}(E)}{k_{\text{tot}_c}(E) - k_{\text{tot}_l}(E)}\left(e^{-k_{\text{tot}_l}(E)t} - e^{-k_{\text{tot}_c}(E)t}\right) \qquad (12)$$

if $[\mathcal{B}] = 0$ at time $t = 0$. The rate equation for formation of $\mathcal{C}$, i.e., monocyclic $C_{10}$ in our case, is given by

$$\frac{d[\mathcal{C}]}{dt} = k_{\text{VAD}_c}(E)[\mathcal{B}] = f_0(E)\frac{k_{\text{iso}}(E)k_{\text{VAD}_c}(E)}{k_{\text{tot}_c}(E) - k_{\text{tot}_l}(E)}\left(e^{-k_{\text{tot}_l}(E)t} - e^{-k_{\text{tot}_c}(E)t}\right) \qquad (13)$$

This differential form for the concentration of monocyclic $C_{10}$, when integrated over all energies $E$ above the isomerization barrier $E_B$, gives the instantaneous flux at a particular time $t$. Thus,

$$\mathcal{R}_{\text{iso}_{1,2}}(t) = N_{1,2}\int_{E_B}^{\infty}f_0(E)\frac{k_{\text{iso}}(E)k_{\text{VAD}_c}(E)}{k_{\text{tot}_c}(E) - k_{\text{tot}_l}(E)}\left(e^{-k_{\text{tot}_l}(E)t} - e^{-k_{\text{tot}_c}(E)t}\right)dE \qquad (14)$$

The instantaneous flux of monocyclic $C_{10}$ as a function of time is the neutralization rate upon isomerization from photoexcited linear $C_{10}^-$.

**Data availability**. The data that support the plots within this paper and other findings of this study are available from the corresponding authors upon reasonable request.

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

## Acknowledgements

We would like to dedicate this paper to the memory of the late Professor Dr. Dirk Schwalm. This work would not have been possible without his contributions and insightful comments. The research was supported by the Benoziyo Endowment Fund for the Advancement of Science.

## Author contributions

Experiments and analysis of the experimental data were conducted by K.S. The experiment was designed by K.S. and V.C. The experimental facilities were developed by O.H., M.L.R., and D.Z. Theoretical calculations were performed by K.S., V.C., and M.A.I. The manuscript was written by K.S. and O.H. All authors discussed the results and commented on the paper.

## Additional information

**Competing interests:** The authors declare no competing financial interests.

