## [Peer Review File · Nature Communications]

Reviewer #1 (Remarks to the Author):

General comments and questions

- I found the paper very clear and interesting. The context, experimental setup, and model are clearly described. There is a single and outstanding result, which is of interest to several communities, hence this is suitable for publication in Nat. Com.

- My main general comment concerns the application to astrophysical environments. If the authors want say their results may be relevant to astrochemistry (which I think they should), it should be discussed a little bit more why this is true. The ultraslow isomerization process is likely to be important only in environments where the anionic species C10⁻ dominates over the neutral or cation. In astrophysical environments, the presence of C10⁻ will be set by the competition between formation and destruction mechanisms. Destruction includes photo-induced electron detachment, but also a number of additional processes (like mutual neutralization with ions like C⁺). Formation processes are also numerous, and related to several parameters such as electron affinity and electron density in the medium. Overall, it may well be that the l-C10⁻ species is not present at all in space, while instead l-C10 or l-C10⁺ are, and their isomerization into c-C10 or c-C10⁺ may be much more efficient. What I am trying to say is that, although indeed C10⁻ isomerization is not efficient as demonstrated here, if C10 or C10⁺ are the dominant species in space, then isomerization could be efficient there.

So my question is : have the authors tried to estimate in which type of astrophysical environments C10⁻ could be present ? I believe this was investigated in chemical models of Herbst and collaborators for instance, so it might be worth looking into these models what is predicted for C10⁻.

- My second comment is related to the first one, but perhaps more a question of phrasing. It seems a bit strange to me to say that ultraslow reactions can be important to help understand the formation of species. For instance in the conclusions :

"This process may shed light on the formation mechanism of larger and more complex molecules and clusters in ISM from simpler species..."

The way I see it is rather (and I think is a more general conclusion) : to understand all possible chemical paths that can lead or not to the formation of new species (including large ones like PAHs or C60) requires to understand and characterize all the competing processes and associated molecular parameters (which is what you are doing here). These (e.g. the rate coefficients in Fig. 5) are the *key inputs* in astrochemical models which try to explain or predict the formation of species in space.

Minor comments

l29-30 : please specify a tracer of what when speaking of HNC/HCN (e.g. a tracer of UV vs Xray dominated regions)

l49 ... may shed light on the more general unexplained creation of large carbon-based systems like polycyclic aromatic hydrocarbons (PAH)²⁴ and C60

-> a more detailed astrochemical model for C60 formation is given in Berné Montillaud & Joblin 2015

Some questions of curiosity regarding the models for RF :

- regarding your model to estimate the RF emission (Eq. 3). RF photons have been measured for

C6- recently (Hansen et al. PRL 2016), has your model been benchmarked well against such experimental results ? Is it feasible ?

- how does your model for RF compares to the one of C60- from Andersen et al. 2001 (Eur. Phys. J. D) ?

- my experience with such models is that the rates depend a lot on how well the electronic structure of the species is known (i.e. what goes in $A_j(E_j)$ in Eq. 3). How robust are your conclusions to these values ? My guess would be they are robust since your RF rates are so low compared to VAD, but I would be happy if you could comment on that.

Reviewer #2 (Remarks to the Author):

The authors investigate the laser irradiation of C10 anions (produced from a Cs sputtering source) and subsequent neutral species formation rate by means of a unique and interesting experimental setup. They interpret the rate of neutral generation after a laser irradiation pulse of trapped C10 anions as evidence for a linear chain to monocyclic anion ring isomerization. A model to account for that particular conversion rate is presented. This work could indicate a slow isomerization process for fundamental forms of carbon clusters, which could be of interest in disciplines of chemistry and astrophysics. However, to this reader at least, the most significant results are the experimental observations. Currently, it seems somewhat difficult to conclude that the observed neutral rate is unique, and thus unclear if it can be used to describe the proposed chain to ring conversion that takes place over microseconds. This is an interesting report, but a more systematic and complete set of experiments that compare several cluster anion sizes and with neutral rates reported from the two different detection modes (i.e., MCPs) that are already available with this instrument would certainly help to establish that a unique neutralization event is taking place.

1) It is rather challenging to attribute the observed neutrals in this work to be an isomerization process or potentially even a unique event because there is no experimental comparison to other carbon cluster anions.

For example, in a recent paper by the same authors/group (Rev. Sci. Instrum. 88, 053101, 2017), which is not apparently cited in this work, the C5 anion is investigated with this instrument in a similar setup but the neutrals are detected at the MCP in the linear end. That C5 anion is not expected to exhibit isomerization to a ring, but neutrals are formed over hundreds of microseconds after the laser pulse interaction. Thus, a delayed neutral production event that takes place over hundreds of microseconds, which is not the result of ring to chain isomerization (but rather due to laser-induced autodetachment) is observed. The decay might be slightly more moderate in this submission but it takes over similar time period. Thus, the experimental analysis of carbon cluster anions that cover a range of cluster sizes (such as C8 to C15 or larger) would clarify what the observations can be applied to a slow isomerization.

2) The authors performed an identical experiment with the MCP detector in the linear portion of the instrument and observed neutrals, even if somewhat 'negligible', but it is not shown. That data should be included in a supporting information document, as well as any other supporting data. Moreover, the neutrals observed at both detectors for a given cluster anion size after the laser pulse could help devise a more systematic set of data to draw conclusions.

3) What is the effect on the neutralization processes if the laser pulse is applied at different intervals of time within the 250 ms trapping event? (Again, at detectors at the bent and linear ends)

4) There have been numerous reports that investigated smaller carbon cluster anions and their

conversions from chains to rings through collisions by ion mobility (e.g., Bowers and co-workers; Jarrold and co-workers) and even some laser irradiation reports (e.g., Eberhardt and co-workers). Perhaps these results could also help support the proposed processes in the context of a larger body of carbon cluster work, such as if a linear chain for a particular cluster size is the dominant starting isomeric form of a laser-induced process, etc.

We would like to thank both reviewers for their important and very helpful comments and suggestions. We have used their comments to improve the main text of the paper as well as to add supplementary information as suggested. Our response to the reviewers' comments (in italics) and the details of the corresponding modifications made in the revised manuscript are presented below. The changes are marked by blue text color in the revised manuscript.

Reviewer #1 (Remarks to the Author):

General comments and questions

- I found the paper very clear and interesting. The context, experimental setup, and model are clearly described. There is a single and outstanding result, which is of interest to several communities, hence this is suitable for publication in Nat. Com.

- My main general comment concerns the application to astrophysical environments. If the authors want say their results may be relevant to astrochemistry (which I think they should), it should be discussed a little bit more why this is true. The ultraslow isomerization process is likely to be important only in environments where the anionic species C_{10}^- dominates over the neutral or cation. In astrophysical environments, the presence of C_{10}^- will be set by the competition between formation and destruction mechanisms. Destruction includes photo-induced electron detachment, but also a number of additional processes (like mutual neutralization with ions like C^+). Formation processes are also numerous, and related to several parameters such as electron affinity and electron density in the medium. Overall, it may well be that the $l-C_{10}^-$ species is not present at all in space, while instead $l-C_{10}$ or $l-C_{10}^+$ are, and their isomerization into $c-C_{10}$ or $c-C_{10}^+$ may be much more efficient.

What I am trying to say is that, although indeed C_{10}^- isomerization is not efficient as demonstrated here, if C_{10} or C_{10}^+ are the dominant species in space, then isomerization could be efficient there. So my question is : have the authors tried to estimate in which type of astrophysical environments C_{10}^- could be present ? I believe this was investigated in chemical models of Herbst and collaborators for instance, so it might be worth looking into these models what is predicted for C_{10}^- .

We added a sentence and a citation to a review article (Ref. 24 in the revised manuscript) to the Introduction concerning the role of negative ions in space. At the end of the manuscript we have added a few sentences specifically discussing C_{10}^- and its main contribution to reactions in outer space, particularly in regard to the astrochemical model in Ref. 52 (added to the revised manuscript) which discusses its contribution to the enhancement of neutral abundance.

Changes in the revised manuscript:

- ***Line No.:47-48. Sentence added in Introduction. Reference added.***
- ***Line No.:264-269. Sentences added in Discussions and Conclusion. Reference added.***

- My second comment is related to the first one, but perhaps more a question of phrasing. It seems a bit strange to me to say that ultraslow reactions can be important to help understand the formation of species. For instance in the conclusions :

“This process may shed light on the formation mechanism of larger and more complex molecules and clusters in ISM from simpler species....”

*The way I see it is rather (and I think is a more general conclusion) : to understand all possible chemical paths that can lead or not to the formation of new species (including large ones like PAHs or C60) requires to understand and characterize all the competing processes and associated molecular parameters (which is what you are doing here). These (e.g. the rate coefficients in Fig. 5) are the *key inputs* in astrochemical models which try to explain or predict the formation of species in space.*

We adopted the reviewer’s rephrasing and changed the text accordingly. We changed Fig. 5 in the manuscript to also show the rate coefficients for isomerization and for relaxation from monocyclic C₁₀⁻ without considering any rotational energy shift, which could be useful in astrochemical models.

Changes in the revised manuscript:

- ***Line No.:269-274. Sentences added in Discussions and Conclusion.***
- ***Fig. 5 changed. Sentence added to its legend (line no. 7-8 in the figure legend).***

Minor comments

l29-30: please specify a tracer of what when speaking of HNC/HCN (e.g. a tracer of UV vs Xray dominated regions)

We have changed this part to be more specific: “In the case of ISM, one example of a key isomerization process is HCN ↔ NHC^{15,16}, in which the line ratio of HCN/NHC is used to trace gas temperatures in galaxies and to compare their luminosities.”

Changes in the revised manuscript:

- ***Line No.:29-30. Sentence modified.***

149 ... may shed light on the more general unexplained creation of large carbon-based systems like polycyclic aromatic hydrocarbons (PAH)²⁴ and C₆₀

-> a more detailed astrochemical model for C₆₀ formation is given in Berné Montillaud & Joblin 2015

Yes, there indeed are a few models which can be divided into bottom-up models and top-down models (as in the reference mentioned by the reviewer). Since we cannot say anything about the preferred model, we only say that evolving from a linear to a more complex structure is an important step in building up more complex molecules assuming the bottom-up models.

Some questions of curiosity regarding the models for RF :

- regarding your model to estimate the RF emission (Eq. 3). RF photons have been measured for C₆-recently (Hansen *et al.* PRL 2016), has your model been benchmarked well against such experimental results ? Is it feasible ?

Yes, we have tested our model with data from C₆⁻ (and others); see, for example Refs. 39 and 43 in revised manuscript (were Refs. 38 and 42 in the original manuscript), which are consistent with Hansen *et al.*, PRL 2016 (Ref. 38 in the revised manuscript, was Ref. 37 in the original manuscript).

- how does your model for RF compares to the one of C₆₀- from Andersen *et al.* 2001 (*Eur. Phys. J. D*) ?

Andersen *et al.* in their model also claim that IR emission by itself cannot explain the relatively fast electron emission from excited C₆₀⁻. They claim that electronic transitions should happen. In their paper, they called it a plasmonic transition. In C₆₀⁻ there is a large probability for having a large density of states because of the many degrees of freedom; therefore, their model is fine. In smaller molecules, like C₆⁻ and C₁₀⁻, more specific electronic states need to be considered and hence the RF process is used here.

- my experience with such models is that the rates depend a lot on how well the electronic structure of the species is known (i.e. what goes in $A_j(E_j)$ in Eq. 3). How robust are your conclusions to these values ? My guess would be they are robust since your RF rates are so low compared to VAD, but I would be happy if you could comment on that.

The model is very robust due to the reason that the reviewer mentioned. Hence, we have preferred to use published values for the electronic states and oscillator strengths (Ref. 49 in the revised manuscript, was Ref. 48 in the original manuscript) to determine $A_j(E_j)$.

Reviewer #2 (Remarks to the Author):

The authors investigate the laser irradiation of C10 anions (produced from a Cs sputtering source) and subsequent neutral species formation rate by means of a unique and interesting experimental setup. They interpret the rate of neutral generation after a laser irradiation pulse of trapped C10 anions as evidence for a linear chain to monocyclic anion ring isomerization. A model to account for that particular conversion rate is presented. This work could indicate a slow isomerization process for fundamental forms of carbon clusters, which could be of interest in disciplines of chemistry and astrophysics. However, to this reader at least, the most significant results are the experimental observations. Currently, it seems somewhat difficult to conclude that the observed neutral rate is unique, and thus unclear if can be used to describe the proposed chain to ring conversion that takes place over microseconds. This is an interesting report, but a more systematic and complete set of experiments that compare several cluster anion sizes and with neutral rates reported from the two different detection modes (i.e., MCPs) that are already available with this instrument would certainly help to establish that a unique neutralization event is taking place.

1) It is rather challenging to attribute the observed neutrals in this work to be an isomerization process or potentially even a unique event because there is no experimental comparison to other carbon cluster anions.

For example, in a recent paper by the same authors/group (Rev. Sci. Instrum. 88, 053101, 2017), which is not apparently cited in this work, the C5 anion is investigated with this instrument in a similar setup but the neutrals are detected at the MCP in the linear end. That C5 anion is not expected to exhibit isomerization to a ring, but neutrals are formed over hundreds of microseconds after the laser pulse interaction. Thus, a delayed neutral production event that takes place over hundreds of microseconds, which is not the result of ring to chain isomerization (but rather due to laser-induced autodetachment) is observed. The decay might be slightly more moderate in this submission but it takes over similar time period. Thus, the experimental analysis of carbon cluster anions that cover a range of cluster sizes (such as C8 to C15 or larger) would clarify what that the observations can be applied to a slow isomerization.

Yes, indeed other carbon clusters (as well many other molecules) do show delayed emission. Here we also cite our work with C_6^- (Refs. 39 and 43 in the revised manuscript; were Refs. 38 and 42 in the original manuscript) and we have data for other carbon clusters, which are beyond the scope of this work that discusses the neutralization processes in photoexcited C_{10}^- . The neutralization rate of photoexcited anionic species depends on its internal dynamics (which is related to the level densities that are unique for each species) and also on the photoexcitation energy. The main difference (which may not have been clear enough in the original text) is that in all other cases the excitation energy was close to the adiabatic electron affinity of the relevant cluster. However, in our experiments, we intentionally tuned the excitation energy to be **much below** the electron affinity of the linear C_{10}^- . Therefore energy-wise, only cyclic C_{10}^- could be neutralized either directly or following isomerization from linear C_{10}^- . Direct electron emission

from cyclic C_{10}^- at photon energies much higher than its adiabatic electron affinity (which is the case in our study) would be fast and would not contribute to the long neutralization rates we observed. We measured the number of neutral species that are promptly produced after the laser interaction, and compared it to the neutral species formed after a delay, and found it to be negligible, indicating the absence of any significant amount of cyclic C_{10}^- in the trapped ion beam (see also the discussion in the Supplementary Information). So, the only possibility remaining to observe neutral species in our experiments is via electron emission from cyclic C_{10}^- after isomerization from linear C_{10}^- . Another indication that the ultraslow neutralization rate is due to isomerization is the fact that this ultraslow neutralization only starts to take place when the cluster is excited to close to the isomerization barrier energy so that the total internal energy of the system is above the barrier. We changed the text to make this explanation more clear.

Changes in the revised manuscript:

- ***Line No.:104-113. Sentences added.***

2) *The authors performed an identical experiment with the MCP detector in the linear portion of the instrument and observed neutrals, even if somewhat 'negligible', but it is not shown. That data should be included in a supporting information document, as well as any other supporting data. Moreover, the neutrals observed at both detectors for a given cluster anion size after the laser pulse could help devise a more systematic set of data to draw conclusions.*

We have included this data in a Supplementary Information document along with relevant explanations, as suggested by the reviewer.

The neutralization rate for delayed neutrals produced from photoexcited C_{10}^- will be the same irrespective of the detector employed (either at the bent or linear ends of the trap) to measure the neutral species. As the detector in the linear end of the trap has a central hole to admit the laser into the trap, there is a significant loss in the number of neutrals that we can detect, resulting in much poorer statistics, especially for long time durations after laser excitation. This is why we employ the detector in the bent section of the EIBT for our studies.

Changes in the revised manuscript:

- ***Lines 122-123. Supplementary information cited.***

3) *What is the effect on the neutralization processes if the laser pulse is applied at different intervals of time within the 250 ms trapping event? (Again, at detectors at the bent and linear ends)*

This data was also included in the Supplementary Information document. At shorter trapping times, the internal energy of the cluster is higher, and this energy decreases with time by radiative relaxation mechanisms. Thus, the total internal energy of C_{10}^- is higher when it is

photoexcited at shorter trapping times, resulting in a faster isomerization rate and higher neutralization probability. Since the internal energy decreases with time, slower neutralization rates are observed when C_{10}^- is photoexcited after longer trapping times.

Changes in the revised manuscript:

- ***Lines 76-77. Supplementary information cited.***

4) There have been numerous reports that investigated smaller carbon cluster anions and their conversions from chains to rings through collisions by ion mobility (e.g., Bowers and co-workers; Jarrold and co-workers) and even some laser irradiation reports (e.g., Eberhardt and co-workers). Perhaps these results could also help support the proposed processes in the context of a larger body of carbon cluster work, such as if a linear chain for a particular cluster size is the dominant starting isomeric form of a laser-induced process, etc.

Yes, indeed there are many studies of the isomers of anionic carbon clusters and their relative abundances. Some of us have also studied it using a sputter source (D. Zajfman *et al.*, *Science* **258**, 1129 (1992), *Zeitschrift für Physik D* **26**, 343 (1993)). It can be concluded that the isomeric ratios are strongly dependent on the ion source type and its conditions. An old summary for C_{10}^- can be found in the review article of Orden *et al.* (*Chem. Rev.*, **98**, 2313 (1998)). Therefore, it is crucial to study the actual population after long trapping time, as presented in the Supplementary Information document.

We hope you will find the revised version of our manuscript suitable for publication in *Nature Communications*.

Reviewer #1 (Remarks to the Author):

I thank the authors for their reply to my comments and for the details they provided for my interest. I do not have more comments to make.

Reviewer #2 (Remarks to the Author):

The clarifications in the revision were helpful and have expanded the previous points in the original submission. However, in the absence of comparative experimental observations under parameters (and with other carbon clusters) that are relevant to this manuscript, it is still not clear if this work reports "an unusually long neutralization" process, which is a major claim of this manuscript (that is, the claim of an "ultraslow isomerization"). The rationale for why C10(-) is likely linear before the laser interaction is reasonable, as is the discussion and results of the model for isomerization of C10(-). The work simply lacks a sort of experimental validation. Nonetheless, it is still of interest for its contributions with respect to the model for isomerization of C10(-) and may potentially spur further investigations into possible isomerization of larger carbon clusters under particular gas-phase environments.

We would like to thank both reviewers for their insightful comments and suggestions which enabled us to improve the manuscript. Our response to the reviewers' comments is presented below.

Reviewer #1 (Remarks to the Author):

I thank the authors for their reply to my comments and for the details they provided for my interest. I do not have more comments to make.

We are thankful to the reviewer for his comments and suggestions.

Reviewer #2 (Remarks to the Author):

The clarifications in the revision were helpful and have expanded the previous points in the original submission. However, in the absence of comparative experimental observations under parameters (and with other carbon clusters) that are relevant to this manuscript, it is still not clear if this work reports “an unusually long neutralization” process, which is a major claim of this manuscript (that is, the claim of an “ultraslow isomerization”). The rationale for why C10(-) is likely linear before the laser interaction is reasonable, as is the discussion and results of the model for isomerization of C10(-). The work simply lacks a sort of experimental validation. Nonetheless, it is still of interest for its contributions with respect to the model for isomerization of C10(-) and may potentially spur further investigations into possible isomerization of larger carbon clusters under particular gas-phase environments.

We do have comparative observations within the C_{10}^- cluster. In our experiments, we have varied the laser wavelength and it is clearly seen in Fig. 4a that with photons of 2.48 eV, there is hardly any neutral production in the long time range and all the neutrals are produced in a relatively short time after the laser excitation. However, if one just increases the photon energy to 2.58 eV (close to the isomerization barrier energy) there is neutral production for a significantly long time. In any previous measurement by us or other groups (see below), any increase in the energy of the photons implies shorter lifetime of the neutral decay – in contrast to the experimental results reported in this manuscript. Therefore, one can conclude that a new neutralization channel is opened. Now the question is what is this new channel? Since, the initial population comprises mostly the linear isomer, the photon energies are not sufficient to neutralize the linear

isomer. The only energetically allowed option to become neutral after the photon absorption is to end up as a neutral cyclic isomer. Now there is a question, what is the slow process? Is it the isomerization process from the linear negative cluster to the cyclic one, or is the isomerization process fast but the vibrational autodetachment (VAD) from cyclic anion slow? Again, the energy criteria dictate that the VAD from cyclic anion is a very fast process at these photon energies. Therefore, the only option in this photon energy range is that the isomerization is the slow process. All these conclusions can be deduced just from the energy level diagram (Fig. 3) and no model is needed. However, in order to explain the rates in a more quantitative way, we are using the statistical model and a nice match to the experimental data is seen. We also put in some additional data of neutral production for photoexcitation after various trapping times that also show the same behavior and agree with the statistical model (see Supplementary Information).

A comparison with other carbon clusters (some of which we have already published) is not directly relevant since each carbon cluster has a different level diagram. For example, in C_6^- the major neutralization process is by the RF process which is about 2 orders of magnitude faster than the VAD process (see refs. 39 and 43). On the other hand, C_5^- has much slower neutral production due to the lack of low lying electronic levels that are needed for the RF process. Similar behavior can be found in the case of C_6^- and C_6H^- [G. Ito *et al.*, PRL **112**, 183001 (2014)], or in the case of C_4^- [Kono *et al.*, Phys. Chem. Chem. Phys. **17**, 24732 (2015)]. Therefore, it can be concluded that the neutralization rate of each cluster by itself depends on its exact level diagram and may have different relaxation rates. It does not mean that slow isomerization cannot be present in other carbon clusters (as well as other molecules), but one cannot use the results of one cluster to predict the behavior of another one without detailed knowledge about its level diagram. In addition, it is not claimed here that such a behavior is unique to C_{10}^- ; on the contrary, it might be that such slow isomerization process is a very common process in larger molecules. We only claim that this was not observed before.

In order to avoid confusion we have deleted the word “unusually” in line 129 of the manuscript.